Third molar agenesis in modern humans with and without agenesis of other teeth

Scheiwiller Maya
http://orcid.org/0000-0001-7949-8785 Oeschger Elias S.
http://orcid.org/0000-0002-3180-6272 Gkantidis Nikolaos nikosgant@yahoo.gr
Department of Orthodontics and Dentofacial Orthopedics, University of Bern , Bern , Switzerland
Eliades Theodore
Electronic publication date: 2020 Nov 17
Publication date: 2020
Volume: 8
Electronic Location ID: e10367
Received 2020 Jul 3; Accepted 2020 Oct 24
Copyright: © 2020 Scheiwiller et al.
Copyright year: 2020
Copyright holder: Scheiwiller et al.
License: This is an open access article distributed under the terms of the Creative Commons Attribution License, which permits unrestricted use, distribution, reproduction and adaptation in any medium and for any purpose provided that it is properly attributed. For attribution, the original author(s), title, publication source (PeerJ) and either DOI or URL of the article must be cited.
License URL: https://creativecommons.org/licenses/by/4.0/

Keywords: Permanent dentition, Tooth agenesis, Third molars, Congenital abnormalities, Hypodontia, Non-syndromic, Evolution, Development

Funding: The authors received no funding for this work.

==============================
Background

The number of teeth in the human dentition is of interest both from developmental and evolutionary aspects. The present case-control study focused on the formation of third molars in modern humans aiming to shed more light on the most variable tooth class in the dentition.

Materials and Methods

For this reason, we investigated third molar formation in a sample of 303 individuals with agenesis of teeth other than third molars (agenesis group) and compared it to a sex and age matched control group of 303 individuals without agenesis of teeth other than third molars.

Results

The prevalence of third molar agenesis in the agenesis group was 50.8%, which is significantly higher than the 20.5% in the control group (p < 0.001). The chance of a missing third molar in the agenesis group was increased by 38.3% (p < 0.001), after controlling for the agenesis in other teeth factor. When considering the amount of missing third molars per individual, a clear tendency towards more missing third molars was evident in the agenesis group compared to the control group. The frequency of bilaterally missing third molars in the agenesis group was 29% in the maxilla, as well as in the mandible, which is about three times higher than the frequency of unilaterally missing third molars (p < 0.001). In the control group, bilaterally missing third molars occurred in 8.6% in the maxilla and 8.9% in the mandible.

Conclusion

The present results indicate that genetic factors involved in tooth agenesis affect also the dentition as a whole. Furthermore, the third molars are more vulnerable to factors involved in agenesis of other teeth and they are more often affected as a whole. These findings seem to be associated with the evolutionary trend in humans towards reduced molar number.

Introduction

Tooth agenesis is the congenital absence of one or more teeth. In the primary dentition, the prevalence ranges between 0.1% and 0.2%. However, in the permanent dentition tooth agenesis is prevalent in 6.4% of the overall population, with similar occurrence in the two jaws (Khalaf et al., 2014). There is a large variation between different population groups and studies (Khalaf et al., 2014).

Tooth agenesis studies generally exclude third molars, due to the high frequency of their absence (Khalaf et al., 2014). Agenesis of third molars is more or less considered a physiologic finding or an evolutionary adaptation of the dentition rather than a developmental disturbance (Koussoulakou, Margaritis & Koussoulakos, 2009). The third molar is the last tooth to develop in the dentition and is characterized by the variability in time of formation and by its diversity in presence or absence (Banks, 1934; Celikoglu et al., 2010). The worldwide average of third molar agenesis is 22.6%, with Asian populations showing the highest rate of 29.7% (Carter & Worthington, 2015).

A wide range of studies shows that the agenesis of third molars correlates with the number of other teeth in the dentition. According to Garn, Lewis & Vicinus (1962), the chance of another tooth to be missing is raised thirteen-fold if at least one-third molar is missing. More recent studies point in the same direction, though with much reduced effect sizes (Bredy, Erbring & Hubenthal, 1991; Celikoglu, Bayram & Nur, 2011; Endo et al., 2015). Endo et al. (2013) reported a significant association between missing third molars and bilateral agenesis of other teeth. Other researchers focused on the agenesis of specific teeth and third molar agenesis (Abe, Endo & Shimooka, 2010; Garib et al., 2010; Garib, Peck & Gomes, 2009).

So far, various studies have investigated the association between missing third molars and agenesis of other teeth, but on limited tooth agenesis samples. Furthermore, most relevant studies tested Asian populations. Thus, we performed a study in a large sample of European subjects, aiming to investigate third molar formation in individuals with and without agenesis of other teeth. To obtain a robust sample, we selected a large number of individuals with agenesis of teeth other than third molars and compared it to a matched group without agenesis of teeth other than third molars. The current approach offers the opportunity to assess previously tested, but also novel questions, relevant to the study hypothesis, with adequate sample sizes. The primary null hypothesis was that there is no difference in third molar agenesis patterns between individuals who have agenesis in teeth other than third molars, and those who do not.

Materials and Methods

In this case-control study, we followed the STROBE guidelines for reporting observational studies (Von Elm et al., 2008).

The ethical approval was provided by the Ethics Commission of the Canton of Bern, Switzerland (Project-ID: 2018-01340) and the Research Committee of the School of Dentistry, National and Kapodistrian University of Athens, Greece (Project-ID: 281, 2/9/2016). The need for informed consent was waived for part of the sample and was obtained for the rest.

Study sample

Consecutive orthodontic records of various time periods within a 12-year period (2006–2018, depending on the place of sample collection) were searched for eligible subjects at the following clinics: (a) University of Bern, Switzerland (b) University of Athens, Greece, (c) two private practices in Athens and two in Thessaloniki, Greece, and (d) one private practice in Biel, Switzerland. Sample collection was performed at the place of data generation by colleagues who were blinded to the aim of this study.

The sample was collected based on the following inclusion criteria:

• Individuals with an age between 12.5 and 40 years

• Individuals with and without agenesis of teeth other than third molars for the agenesis and the control group, respectively

• European ancestry

• No syndromes, systemic diseases or other defects that affect the craniofacial complex development, as reported in the subjects’ medical records

• Adequate quality panoramic radiographs for identification of missing teeth (Fig. S1)

• No individuals where the cause of missing teeth was unclear

• No individuals where the presence or absence of teeth could not be confirmed

The minimum age limit of 12.5 years was determined according to previous studies that evaluated the correlation between chronological age and the degree of third molar mineralization (Caldas et al., 2011; De Oliveira et al., 2012; Karataş et al., 2013; Soares et al., 2015; Zandi et al., 2015). They showed that in 95% of cases, Demirjian’s stage A was observed at the age of 12.5 or younger, which means that the mineralization of third molar crowns has already started and is clearly visible on the panoramic radiographs.

Finally, the panoramic radiographs of 303 individuals with agenesis of teeth other than third molars (agenesis group) were selected from a large orthodontic sample of approximately 10,000 individuals, based on availability. A control group of 303 individuals without agenesis of teeth other than third molars, matched for age (within 6 months) and sex was formed from the same archives. All other inclusion criteria for the control group, were the same as mentioned above for the agenesis group.

Data extraction

After reviewing the orthodontic files (medical and dental history, intraoral and extraoral photos, radiographs) at the place of sample collection, the relevant data were recorded in an Excel sheet (Microsoft Excel; Microsoft Corporation, Redmond, WA, USA) in a standardized manner. To identify tooth agenesis, the panoramic radiographs were digitized and viewed on screen. A single researcher (M.S.) performed the data extraction procedure of the entire sample in terms of missing teeth, and repeated it for 40 randomly selected subjects (https://www.random.org/) following a 1-month washout period. In case of disagreement, the radiographs were controlled by all authors and a consensus was reached.

To record tooth agenesis patterns, the TAC system was used (Van Wijk & Tan, 2006). This system assigns a binary value to each tooth providing a unique numeric value for each pattern. Each dental quadrant is analyzed separately, and thus, the combined values assigned to each of the quadrants (q1, q2, q3, and q4) represent a unique tooth agenesis pattern (Van Wijk & Tan, 2006).

Statistical analysis

All statistical analyses were conducted with SPSS software (IBM SPSS Statistics for Windows, Version 23.0; IBM Corp, Armonk, NY, USA). Descriptive statistics were also calculated through the TAC Data Analysis Tool (http://www.toothagenesiscode.com/, last accessed 15 May 2019). Intra-rater agreement was evaluated through the percentage of different patterns identified in the two repeated assessments. The two-tailed Pearson’s Chi square test was used to assess differences in the frequencies observed in the control and the agenesis samples. The Spearman’s correlation coefficient was used to investigate the relation of the number of agenesis of teeth other than third molars to the number of third molar agenesis, overall, as well as within quadrants.

Results

Method error

The intra-rater agreement between repeated tooth agenesis pattern identification was 97.5%.

Agenesis group without considering third molars

In the 303 individuals (170 females, 133 males) of the agenesis sample, in total 799 teeth, other than third molars, were congenitally missing. In 38.6% of the sample one tooth, in 33.3% two, and in 7.9% three teeth were missing (Table S1). The incidence for missing teeth in the maxilla was 57.1%, compared to 68.6% in the mandible (p = 0.079). The most common missing tooth was the mandibular second premolar (29.3%), followed by the maxillary lateral incisor (21.0%), and the maxillary second premolar (14.0%; Table 1).

Table 1 Distribution of missing teeth across quadrant and tooth number.

Tooth number*	Upper right (%)	Upper left (%)	Lower right (%)	Lower left (%)	Total (%)	
Agenesis group						
1	3 (0.25)	2 (0.2)	29 (2.4)	30 (2.5)	64 (5.3)	
2	85 (7)	83 (6.8)	15 (1.2)	18 (1.5)	201 (16.5)	
3	14 (1.2)	13 (1.1)	5 (0.4)	4 (0.3)	36 (3)	
4	20 (1.6)	21 (1.7)	15 (1.2)	15 (1.2)	71 (5.8)	
5	60 (4.9)	52 (4.3)	117 (9.6)	117 (9.6)	346 (28.4)	
6	5 (0.4)	4 (0.3)	9 (0.7)	7 (0.6)	25 (2.1)	
7	13 (1.1)	14 (1.2)	15 (1.2)	14 (1.2)	56 (4.6)	
8	101 (8.3)	105 (8.6)	104 (8.5)	108 (8.9)	418 (34.3)	
Total	301 (24.7)	294 (24.2)	309 (25.4)	313 (25.7)	1217 (100)	
Control group						
8	37 (25.7)	32 (22.2)	39 (27.1)	36 (25)	144 (100)	
Note:

* Tooth number 1 stands for all central incisors (11, 21, 31, 41) and so on. For example, Upper right 1 is tooth 11 in the FDI system.

Regarding tooth symmetry, the teeth that were most often symmetrically missing in the maxilla were the lateral incisors (19.5%) followed by the second premolars (14.2%). In the mandible, the second premolars were missing bilaterally in 25.4%, followed by the central incisors in 7.9% (Table 2).

Table 2 Frequency tables that show single tooth agenesis and the prevalence of right sided, left sided or bilateral agenesis in the whole sample (n = 606).

Maxilla	
Tooth number*	Present bilaterally (%)	Missing right side (q1) (%)	Missing left side (q2) (%)	Missing unilaterally (%)	Missing bilaterally (%)	
Agenesis group						
1	300 (99.0)	1 (0.3)	0 (0.0)	1 (0.3)	2 (0.7)	
2	194 (64.0)	26 (8.6)	24 (7.9)	50 (16.5)	59 (19.5)	
3	286 (94.4)	4 (1.5)	3 (1.0)	7 (2.5)	10 (3.3)	
4	277 (91.4)	5 (1.7)	6 (2.0)	11 (3.7)	15 (5.0)	
5	234 (77.2)	17 (5.6)	9 (3.0)	26 (8.6)	43 (14.2)	
6	297 (98.0)	2 (0.7)	1 (0.3)	3 (1.0)	3 (1.0)	
7	287 (94.7)	2 (0.7)	3 (1.0)	5 (1.7)	11 (3.6)	
8	185 (61.1)	13 (4.3)	17 (5.6)	30 (9.9)	88 (29.0)	
Control group						
8	260 (85.8)	11 (3.6)	6 (2.0)	17 (5.6)	26 (8.6)	
Mandible	
Tooth number	Present bilaterally (%)	Missing right side (q4) (%)	Missing left side (q3) (%)	Missing unilaterally (%)	Missing bilaterally (%)	
Agenesis group						
1	268 (88.4)	5 (1.7)	6 (2.0)	11 (3.7)	24 (7.9)	
2	281 (92.7)	4 (1.3)	7 (2.3)	11 (3.6)	11 (3.6)	
3	297 (98.0)	2 (0.7)	1 (0.3)	3 (1.0)	3 (1.0)	
4	284 (93.7)	4 (1.3)	4 (1.3)	8 (2.6)	11 (3.6)	
5	146 (48.2)	40 (13.2)	40 (13.2)	80 (26.4)	77 (25.4)	
6	292 (96.4)	4 (1.3)	2 (0.7)	6 (2.0)	5 (1.7)	
7	283 (93.4)	6 (2.0)	5 (1.7)	11 (3.7)	9 (3.0)	
8	179 (59.1)	16 (5.3)	20 (6.6)	36 (11.9)	88 (29.0)	
Control group						
8	255 (84.2)	12 (4.0)	9 (3.0)	21 (7.0)	27 (8.9)	
Note:

* Tooth number 1 stands for all central incisors (11, 21, 31, 41) and so on. For example, Upper right 1 is tooth 11 in the FDI system.

The most common agenesis patterns in the maxilla were bilaterally missing lateral incisors (23.1%), followed by bilaterally missing second premolars (12.7%). In the mandible, the most common patterns were bilateral agenesis of second premolars in 27.9%, followed by unilateral agenesis of the right second premolar (17.3%). In the whole dentition, bilateral agenesis of maxillary lateral incisors occurred most often (11.2%), followed by bilateral agenesis of mandibular second premolars (10.2%; Table 3).

Table 3 Most common tooth agenesis patterns in the agenesis group excluding third molars.

	Frequency (%)	Missing teeth		Frequency (%)	Missing teeth	
Maxilla	Mandible	
1	40/173 (23.1)	12, 22	1	58/208 (27.9)	35, 45	
2	22/173 (12.7)	15, 25	2	36/208 (17.3)	45	
3	21/173 (12.1)	12	3	34/208 (16.3)	35	
4	20/173 (11.6)	22	4	10/208 (4.8)	31, 41	
5	11/173 (6.4)	15	5	5/208 (2.4)	34, 35, 44, 45
or
32, 42	
Overall	114/173 (65.9)		Overall	143/208 (68.8)		
Whole dentition	
1	34/303 (11.2)	12, 22	
2	31/303 (10.2)	35, 45	
3	29/303 (9.6)	45	
4	27/303 (9.0)	35	
5	18/303 (6.0)	22	
Overall	139/303 (45.9)		

Third molar agenesis in the agenesis and the control group

The prevalence of third molar agenesis in the agenesis group was 50.8%, which is significantly larger than the 20.5% in the control group (p < 0.001). A total of 418 third molars were congenitally missing in the agenesis group (n = 303) compared to 144 in the control group (n = 303). If the probability of third molar agenesis in the agenesis group was equal to that of teeth other than third molars, this would increase the value of 144 missing third molars, observed in the control group, by 114. Consequently, 258 missing third molars would have been expected in the agenesis group. This value is significantly lower than the actual value observed (418; p < 0.001). Thus, the chance of a missing third molar in the agenesis group is increased by 38.3%, compared to controls.

In the agenesis group, there was a significant, though weak correlation, of the total number of other missing teeth to the total number of missing third molars (rho = 0.31, p < 0.001). Similarly, very weak correlations were identified when third molar agenesis was correlated to the number of other tooth agenesis within quadrants (Q1: rho = 0.16, p = 0.006; Q2: rho = 0.14, p = 0.015; Q3: rho = 0.20, p = 0.001; Q4: rho = 0.29, p = 0.001).

The frequency of bilaterally missing third molars in the agenesis group was 29% in the maxilla, as well as in the mandible. This is about three times higher than the frequency of unilaterally missing third molars (maxilla: 9.9%, p < 0.001, mandible: 11.9%, p < 0.001; Table 2). The ratio of bilateral to unilateral third molar agenesis was significantly higher in the agenesis group compared to the control group (maxilla: 2.93 vs. 1.53, respectively, p < 0.001; mandible: 2.44 vs. 1.29, respectively, p < 0.001; Table 2).

In the tooth agenesis group, symmetrical third molar agenesis occurred in a similar manner within jaws (29% within each jaw), between jaws (right side: 24%, left side: 24%), or crossed quadrant (q1 vs. q3: 22.1%; q2 vs. q4: 24.4%) (p > 0.05; Table 4). The same was true for the control groups (p > 0.05; Table 4), though the prevalence of all respective symmetrical patterns was much lower (range: 6.6–8.9%, p < 0.001).

Table 4 Symmetry of tooth agenesis patterns.

	Comparison		Symmetry I (%)	Symmetry II (%)	
		No 3rd—Agenesis	26.7	46.8	
Maxilla	Right vs. left side	3rd—Agenesis	29.0	74.6	
		3rd—Control	8.6	60.5	
		No 3rd—Agenesis	30.0	43.8	
Mandible	Right vs. left side	3rd—Agenesis	29.0	71.0	
		3rd—Control	8.9	56.3	
		No 3rd—Agenesis	6.9	8.8	
Left side	Upper left vs. lower left	3rd—Agenesis	24.1	52.1	
		3rd—Control	6.6	41.7	
		No 3rd—Agenesis	7.9	9.8	
Right side	Upper right vs. lower right	3rd—Agenesis	24.4	56.5	
		3rd—Control	8.6	52.0	
		No 3rd—Agenesis	6.9	8.71	
Crossed q1 vs. q3	Upper right vs. lower left	3rd—Agenesis	22.1	47.2	
		3rd—Control	6.6	37.7	
		No 3rd—Agenesis	7.6	9.5	
Crossed q2 vs. q4	Upper left vs. lower right	3rd—Agenesis	24.4	54.8	
		3rd—Control	6.6	39.2	
Note:

Symmetry I: percentage relative to the whole sample (n = 303) without considering the patterns of no missing teeth as symmetrical. Symmetry II: percentage relative to subsample of subjects with missing teeth in the respective area (i.e., maxilla, mandible etc.).

In both groups, there was no statistically significant difference between the number of missing third molars in the different quadrants (Chi square test, p > 0.05; Table 1). The agenesis group differed significantly from the control group in the distribution of the number of missing third molars (p < 0.001). There is a clear tendency towards more missing third molars in the agenesis group compared to the controls. The agenesis group has 1.55, 2.14, 3.80, and 3.48 times higher possibility of having one, two, three, or four missing third molars respectively, when compared to the control group (Fig. 1).

Figure 1 Distribution of individuals with different number of missing third molars (x-axis) in the agenesis and the control group.

Table S2 shows the most common patterns of tooth agenesis in the agenesis group, including third molars. In the maxilla, the lateral incisors were most commonly missing in 14.1%, followed by bilaterally missing third molars in 13.6% of the sample. In the mandible, the second premolars were most commonly missing bilaterally in 12.8% of the sample, followed by unilateral second premolar agenesis.

Table S3 shows the most common patterns of third molar agenesis in control subjects where agenesis was observed. In the maxilla, as well as in the mandible, bilateral third molar agenesis was the most common pattern (60.5% and 56.3%, respectively). In the entire dentition, the most common pattern was the four missing third molars (17.4%), followed by bilateral third molar agenesis in the mandible, in 14.5%. Table S4 shows the most common patterns of third molar agenesis in the agenesis group, where third molar agenesis was observed. In this group also, bilateral third molar agenesis was the most common pattern within jaws (74.6% and 71.0%, in the maxilla and the mandible, respectively). Furthermore, in the entire dentition, the most common pattern was also in this group the four missing third molars (38.3%), followed by bilateral third molar agenesis in the mandible (12.3%).

Discussion

The purpose of this study was, to explore the patterns of third molar agenesis in a large sample of modern European subjects with and without agenesis of other teeth. The prevalence of third molar agenesis in the agenesis group was 50.8%, which is about 2.5 times higher than in the control group. In the agenesis group, there was a weak correlation of the number of agenesis of other teeth with the number of third molar agenesis within individuals, as well as very weak correlations of third molar agenesis to the number of agenesis of other teeth within quadrants. When considering the percentages of the amount of missing third molars per individual in the control and the agenesis group, there was a clear tendency towards more missing third molars in the agenesis group. The frequency for bilaterally missing third molars in the agenesis group was about three times higher than the frequency of unilateral absence. The ratio of bilateral to unilateral third molar agenesis was also significantly higher in the agenesis group, compared to the control group.

It could also be useful to note, that based on our findings, in young patients with severe tooth agenesis, the clinician should expect that probably the third molars will also be missing. This should be considered in the treatment planning of severe tooth agenesis cases, which are usually complex and require a multidisciplinary approach.

Our methodology differs from all previous studies, in terms that we investigated the patterns of third molar agenesis in a large sample with agenesis of other teeth. To our knowledge, all the existing studies divided their groups according to third molar agenesis. Through the latter approach, only a small percentage of the subsequent subsamples had agenesis in teeth other than third molars, and thus, these groups did not have adequate or comparable size to the control groups. Our study tested a large agenesis sample of 303 agenesis individuals, as well as 303 controls, selected out of a total of around 10,000 records. This allowed for findings that are presented for the first time in the literature, such as those related to symmetry or to occurrences within quadrants. Furthermore, the groups were matched for sex and age, accounting for any confounding effects of these factors. For younger individuals, these might be related to the differences in dental maturity between sexes of the same chronological age and for older individuals, to the etiology of tooth absence. Furthermore, studies on tooth agenesis (Khalaf et al., 2014), as well as on third molar agenesis (Carter & Worthington, 2015), agree in the higher prevalence of agenesis in females than in males. Differences between sexes were not investigated here, since this was beyond the scope of the present study. A potential effect of the sex factor on the outcomes is not expected because the sample was matched for sex.

The age range that we considered was limited from 12.5 to 40 years old. The minimum limit was defined according to various longitudinal studies that showed the correlation between chronological age and the degree of third molar mineralization using Demirjian’s developmental stages. This classification has been widely used and tested to facilitate age estimation. Therefore, the choice of this age limit is considered to be appropriate for our purpose (Caldas et al., 2011; De Oliveira et al., 2012; Karataş et al., 2013; Soares et al., 2015; Zandi et al., 2015). The upper age limit of 40 years was chosen to avoid false positive results due to extraction or tooth loss due to other reasons that could have been registered as agenesis.

We found a prevalence of 50.8% for third molar agenesis in the agenesis group compared to 20.5% in the control group. According to a recent meta-analysis (Carter & Worthington, 2015), the worldwide average of third molar agenesis is 22.6% (21.6% for Europeans), confirming the validity of our control group. Our results clearly demonstrated that in individuals with agenesis of other teeth, the prevalence of third molar agenesis is higher. This points in the same direction with previous studies that showed an increased prevalence of agenesis of other teeth in individuals with third molar agenesis (Bredy, Erbring & Hubenthal, 1991; Celikoglu, Bayram & Nur, 2011; Endo et al., 2015).

In our control group, the sequence of the number of missing third molars was similar to that of Carter and Worthington (Carter & Worthington, 2015) that showed the highest prevalence for one missing third molar, followed by two, and four missing third molars. However, the most common amount of missing third molars in the agenesis group was four, followed by two and one third molar. This inconsistency is attributed to the different sample composition. The aforementioned meta-analysis tested third molar agenesis in the general population, meaning that individuals with agenesis of other teeth would be limited. The above findings clearly show that the presence of agenesis, in teeth other than third molars, has a considerable effect on third molar agenesis patterns. Especially, the probability to have four missing third molars increases. This suggests that the third molars might be more vulnerable to genetic factors involved in tooth agenesis, as compared to other tooth types. Indeed, this is also supported by the increased number of missing third molars in the agenesis sample compared to that expected by chance. A recent study analyzing data from 172 monozygotic and 112 dizygotic twins concluded that third molar formation is strongly controlled by additive genetic factors, providing further support to our statements (Trakinienė et al., 2018). This concept is in line with the evolutionary trend in humans towards less teeth, and more specifically, less molars (Kavanagh, Evans & Jernvall, 2007). Facial size has also been reduced during evolution (Bastir et al., 2010). Recent evidence showed that the number of teeth that are formed in a dentition is associated with facial size in modern humans. This indicates that a biological mechanism of tooth number reduction that has evolved during time might still be active and continue to regulate the number of teeth and facial size in a coordinated manner (Oeschger et al., 2020). The findings of the present study, along with the high prevalence of third molar agenesis in the population (Carter & Worthington, 2015) suggest that the third molars might be affected to a higher degree from such mechanisms, compared to other teeth in the dentition.

Furthermore, in terms of developmental timing, the third molar is the last tooth in the molar series and also the last tooth to develop in the dentition. Evidence supports that the last tooth in each tooth series shows more often developmental disturbances, including agenesis, thus being more vulnerable to genetic or environmental factors that might be present during development (Townsend et al., 2009; Gkantidis et al., 2017). This might be another contributing factor relevant to the present results. It has also been shown that overall dental development is delayed in patients with tooth agenesis, compared to controls, with a weak correlation between dental developmental stage and number of missing teeth (Lebbe et al., 2017).

In the agenesis group, the prevalence for bilaterally missing third molars was more than three times higher than in the control group, in the maxilla as well as in the mandible. The ratio of bilateral to unilateral third molar agenesis was significantly higher in the agenesis group compared to the control group. The same was true for all types of symmetry. Furthermore, in the agenesis and the control group, the most common third molar agenesis pattern was four missing third molars, followed by bilateral third molar agenesis in the mandible. This is in line with our previous statement that third molars are more susceptible to genetic or epigenetic factors that cause tooth agenesis, and might more possibly be affected as a whole. The above claim is also supported by the increasing possibility for more missing third molars in the agenesis group than in the controls. Furthermore, very weak correlations were identified between other missing teeth and third molar agenesis within quadrants, suggesting that there are no significant genetic effects limited within quadrants.

A limitation of the study could be that the sample was selected from orthodontic practices, meaning that it may not be representative of the general population. For example, it might be evident that the percentage of severe tooth agenesis occurrences is higher in our sample, since it derived from orthodontic patients, including two university centers. However, the study aimed to test the association of third molar formation with the formation of other teeth using a case-control study design. Thus, the study did not aim to represent the general population, but it aimed to test the association of the severity of agenesis of other teeth with third molar formation. Therefore, this is not considered a limitation of the study. On the contrary, it led to adequate number of cases representing the occurrence of severe agenesis (n = 27 cases with 6 or more teeth missing, 8.9% of the agenesis sample). The fact that the sample derived from orthodontic patients, which are thoroughly documented and followed over time, reduces the possibility that certain severe cases might represent non-diagnosed syndromes. Even if limited misdiagnosed cases were present in the sample, our analysis showed very weak correlations of the total number of other missing teeth to the total number of missing third molars. Thus, these cases could have not confounded our findings considerably. Without considering third molars, the present tooth agenesis patterns are comparable to those of other studies presented in the literature (Gkantidis et al., 2017; Khalaf et al., 2014), concerning the most common missing teeth and other tooth agenesis patterns. Regarding other characteristics of an orthodontic population, when considering that malocclusion is endemic in recent years, it is not expected that our sample would highly differ from the general population. Another limitation could be the inclusion of patients up to 40 years old, which might increase the chance to miss information on causes of tooth loss, such as due to extractions. To control for this confounding, according to our inclusion criteria, a case was excluded when the treating doctors judged that the cause for a missing tooth was unclear. Through this approach, misdiagnosis might not be fully excluded, but it was limited considerably, in order not to critically affect the outcomes. Finally, the present results are based only on subjects of the European population, and thus, they have to be confirmed on other ancestries. However, the study sample originated from places where the white European background is highly represented. Thus, we decided to include only white subjects of European ancestry to avoid confounding.

Conclusion

The present study showed that individuals with non-syndromic tooth agenesis in teeth other than third molars show a higher prevalence of third molar agenesis compared to matched control individuals without agenesis of other teeth. There was also a clear tendency towards more missing third molars in the agenesis group. Furthermore, in the agenesis group, the prevalence for bilaterally missing third molars was more than three times higher than in the control group. The ratio of bilateral to unilateral third molar agenesis was also significantly higher. The above findings indicate that the third molars might be more vulnerable to genetic or epigenetic factors involved in agenesis of other teeth and they are often affected as a whole. These findings seem to be associated with the evolutionary trend in humans towards reduced number of teeth.

Supplemental Information

Supplemental Information 1 Distribution of the total number of missing teeth per individual in the agenesis sample, excluding third molars.

Click here for additional data file.

Supplemental Information 2 Most common patterns of tooth agenesis observed in the agenesis group including third molars.

Click here for additional data file.

Supplemental Information 3 Most common patterns of third molar agenesis observed in control individuals.

Click here for additional data file.

Supplemental Information 4 Most common patterns of third molar agenesis observed in the agenesis subjects.

Click here for additional data file.

Supplemental Information 5 Four randomly selected panoramic radiographs from the tested sample.

Click here for additional data file.

Supplemental Information 6 Raw data of tooth agenesis patterns in the agenesis sample provided as defined by the TAC system.

Specific teeth are represented by numbers according to the FDI dental numbering system. 0 value indicates that the tooth is present and 1 value that it is absent due to tooth agenesis.

Click here for additional data file.

Supplemental Information 7 Raw data of tooth agenesis patterns in the control sample provided as defined by the TAC system.

Specific teeth are represented by numbers according to the FDI dental numbering system. 0 value indicates that the tooth is present and 1 value that it is absent due to tooth agenesis.

Click here for additional data file.

Additional Information and Declarations

Competing Interests

Author Contributions

Human Ethics

Data Availability

The authors declare that they have no competing interests.

Maya Scheiwiller conceived and designed the experiments, performed the experiments, analyzed the data, prepared figures and/or tables, authored or reviewed drafts of the paper, and approved the final draft.

Elias S. Oeschger performed the experiments, analyzed the data, prepared figures and/or tables, and approved the final draft.

Nikolaos Gkantidis conceived and designed the experiments, performed the experiments, analyzed the data, prepared figures and/or tables, authored or reviewed drafts of the paper, and approved the final draft.

The following information was supplied relating to ethical approvals (i.e., approving body and any reference numbers):

The ethical approval was provided by the Ethics Commission of the Canton of Bern, Switzerland (Project-ID: 2018-01340) and the Research Committee of the School of Dentistry, National and Kapodistrian University of Athens, Greece (Project-ID: 281, 2/9/2016). The need for informed consent was waived for part of the sample and was obtained for the rest.

The following information was supplied regarding data availability:

Raw data is available in the Supplemental Files.

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
