# Peer review of "Third molar agenesis in modern humans with and without agenesis of other teeth"

_PeerJ, doi:10.7717/peerj.10367_

## Round 0.1 · original submission · Major Revisions

Some of the issues raised by the first reviewer are not manageable without altering parts of the study.

Reviewer 1 ·

Basic reporting

The article is clearly written and meets these criteria. Comments on improving the Tables are included in the general comments.

Experimental design

See general comments

Validity of the findings

Issues with the sample composition, level of analysis and limitations are integrated with the general comments.

Additional comments

This study of tooth agenesis compares third molar agenesis in individuals with agenesis of other teeth (cases) and control individuals with no agenesis of other teeth. The authors are to be commended for their improved study design over many earlier studies on the subject.

The study has problems with aspects of the data collection and samples, analytical methods, presentation of tabular data and inconsistency with dental terminology.
In terms of the data collection, it may be that the case and control selection is not clearly described, but it appears that the individuals were selected from orthodontic patients who had records from the last 12-year period. This is confusing as the sample demographics include individuals up to 40 years of age. Does this mean individuals of age 40 who were orthodontic patients? It seems unlikely that individuals of the older ages would still be regular orthodontic patients or that their records would include information on extractions performed long ago and perhaps by another practitioner, such as premolar removal to facilitate corrective procedures.
Secondly, the authors state that individuals with syndromes were excluded. Yet they included 27 oligodontia cases (almost 9% of the cases) with up to 20 missing teeth (Suppl. Table 1). With 20 missing teeth, the most extreme case had only 8/28 teeth occurring. How is this extreme level of oligodontia justified for inclusion, as non-syndrome oligodontia is extremely rare? The use of these extreme cases seriously compromises the integrity of the results.

The simple investigation of associations among the agenesis patterns is not sufficient for documenting the complexity of agenesis in this study. What are the odds associated with these combinations?

Tables should be simple and easy to understand. They need to be ‘stand alone’, meaning that they can be understood without reference to the text, which should only elaborate upon the data presented in the table. Several in this manuscript are not. Varying use of tooth numbering terminology and descriptors, in the text and tables should be corrected. For example, in Tables 1 and 2 the heading ‘Tooth number’ is not clear and is not consistent with a subsequent table that uses the FDI system (Table 3). The non-standard table terms should be explained as a footnote below the table.
Tables 3 and S2, S3, S4: What does the header “Index’ mean?
Table 4: The header ‘Pattern Symmetry’ seems to be over the wrong column.

Finally, the authors state that their findings are consistent with other studies so the clinical nature of the study is not a true limitation. Unfortunately this point is moot, as agreement with other studies that are also clinical is not relevant to the limitation issues.

Reviewer 2 ·

Basic reporting

The purpose of this study was, to explore the patterns of third molar agenesis in modern European subjects with and without agenesis of other teeth. The topic fits within the scope of PeerJ. The Intro section ends with the null-hypothesis that there is no difference in third molar agenesis patterns between individuals who have agenesis in teeth other than third molars, and those who do not.

1. This is a well-written manuscript that follows the guidelines of the STROBE reporting guideline.

2. The Introduction section is adequate concerning the quoted literature. However, I invite the authors to better explain in the Intro section why a new study was needed – what is the knowledge gap - considering the high-quality systematic review and meta-regression of Carter and Worthington in 2015. Their analysis included 24 studies on Europeans, so this constituted a large sample size for meta-analysis.

3. Figures and tables are relevant but there is some improvement possible.

Experimental design

4. This is a well-conducted study, but the authors do not clearly state the design of the study. Is this a case-control study? Please add.

5. The inclusion criteria for the controls could be described better. It is only stated that a control group of 303 individuals without agenesis of teeth other than third molars, were matched for age and sex, but what about the other criteria as for the cases group?

6. It is not completely clear how the study size was arrived at. The authors state that they took a sample of n=303 with agenesis from about 10.000 cases. But it then means that the incidence for agenesis in this sample is about 3%, which is much lower than reported in literature. This does not comply with statements in the discussion section that the sample has external validity. Could you explain, please?

7. Table 1 only gives numbers and not the percentages, please add.

8. Figure 1, legend in the table. ‘’agenesis’’ It is confusing, please make it ‘’agenesis group’’ and ‘’controls’’

Validity of the findings

9. When considering the percentages of the number of missing third molars per individual in the control and the agenesis group, a tendency was found towards more missing third molars in the agenesis group. Thus is the key finding of the study. The authors only mention ‘’genetic factors’’ which is a very vague term. The discussion could be enhanced by trying to explain this finding from the embryological viewpoint, f.e. the studies of Inger Kjaer in Copenhagen.

10. About the external validity, see my question above.

11. The discussion section could also be broadened by elaborating on the clinical significance of the findings. As a reader I am left with question ‘’OK, so what …?”

12. The raw data are in a text file and rather useless for a check. It is up to the editor to decide if this is sufficient.

---

## Round 0.2 · Major Revisions

Please address the issues raised by the first reviewer.

Reviewer 1 ·

Basic reporting

The article is satisfactory in this regard, but some small corrections of the English is needed.

Experimental design

Some shortcomings with the analysis are discussed in the comments.

Validity of the findings

See comments.

Additional comments

In terms of the data collection, it may be that the case and control selection is not clearly described, but it appears that the individuals were selected from orthodontic patients who had records from the last 12-year period. This is confusing as the sample demographics include individuals up to 40 years of age. Does this mean individuals of age 40 who were orthodontic patients? It seems unlikely that individuals of the older ages would still be regular orthodontic patients or that their records would include information on extractions performed long ago and perhaps by another practitioner, such as premolar removal to facilitate corrective procedures.
Authors’ Response: The reviewer is right that the individuals were selected from orthodontic patients. Various archives were searched to identify pre-treatment records of eligible patients that were obtained the defined 12-year period. It is true that certain patients were also up to 40 years old and this is indeed a clinical occurrence. Of course, this is not the majority of orthodontic patients, but it is only a small part. With a thorough anamnesis and medical history kept in all searched archives, it was possible in most cases to determine the reason of any tooth absence. When the treating doctors judged that the cause for missing teeth was unclear, the case was excluded, according to our eligibility criteria. Through this approach, misdiagnosis might not be fully excluded, but it was limited considerably, in order not to critically affect the outcomes. This consideration has now been added to the limitations of our paper and the issues mentioned just above have been clarified in the methods section.
--The authors have dealt with this issue in a satisfactory manner.

Secondly, the authors state that individuals with syndromes were excluded. Yet they included 27 oligodontia cases (almost 9% of the cases) with up to 20 missing teeth (Suppl. Table 1). With 20 missing teeth, the most extreme case had only 8/28 teeth occurring. How is this extreme level of oligodontia justified for inclusion, as non-syndrome oligodontia is extremely rare? The use of these extreme cases seriously compromises the integrity of the results.
Authors’ Response: As mentioned, our sample consisted orthodontic patients, including two university centers. Thus, the prevalence of oligodontia could be different from the general population, since such severe cases are usually referred at the university and they usually need orthodontic treatment. Regarding syndromes, we based our decisions on the thorough review of the medical and dental history of the patients. Thus, there is always a risk of including a non-diagnosed syndromic patient in our sample. This risk cannot be totally eliminated and this is mentioned in the limitations of our study. The fact that we might have in our sample more oligodontia patients as it might be expected for the general population does not compromise our findings, since the case-control study design, for the hypotheses tested, does not require or did not aim to represent the general population. Thus, for the specific hypothesis tested in the study, including the effect of the number of missing teeth on the third molar formation, it is actually an advantage to have adequate numbers of cases on the different agenesis severity levels. Relevant information is now provided in the limitations section of our manuscript.
--The authors have not dealt with this major bias and design flaw. The problem is not whether oligodontia occurs with greater or lesser frequency in their cases vs. the general population, but their failure to address how these extreme cases (most probably due to syndromes that were supposed to be excluded) may actually bias their results. They should have done an analysis treating them separately to evaluate the impact of their inclusion.

The simple investigation of associations among the agenesis patterns is not sufficient for documenting the complexity of agenesis in this study. What are the odds associated with these combinations?
Authors’ Response: Table 3 and Supplemental Tables 2 and 3 describe the most common tooth agenesis patterns present at the studied sample along with the associated frequencies. These descriptive data provide an overview of the agenesis patterns observed in the sample and along with the other data presented in the study provide a thorough description of the current sample. A more detailed tooth agenesis pattern assessment similar to that previously published by our team is beyond the scope of the current study (Gkantidis N, Katib H, Oeschger E, Karamolegkou M, Topouzelis N, Kanavakis G. Patterns of non-syndromic permanent tooth agenesis in a large orthodontic population. Arch Oral Biol. 2017;79:42-47. doi:10.1016/j.archoralbio.2017.02.020). Furthermore, since the raw data are provided with the submission, future researchers that might be interested in specific patterns can use our data for their purposes.
--This is not a satisfactory response; it is their responsibility to present their findings in an informative, analytical manner and not to leave it up to the reader. Secondly, the article they cite here as containing or pertaining to the requested analysis, is not even referenced with regard to the agenesis patterns in the results.

Finally, the authors state that their findings are consistent with other studies so the clinical nature of the study is not a true limitation. Unfortunately, this point is moot, as agreement with other studies that are also clinical is not relevant to the limitation issues.
Authors’ Response: To clarify this issue, that was also noted by the reviewer in a comment above, we rephrased this paragraph and transferred this information to the limitations section. We hope that following the clarifications provided here and in the revised versions of our manuscript, the reviewer’s concerns on this issue will be adequately addressed.
--Elsewhere the authors have argued that their study is not designed specifically for clinical purposes but ‘ it is more relevant to basic research than to clinical research’, yet they use the clinical basis of their samples to argue for the strength of their results. I think they do not demonstrate an understanding of the underlying biological premises of conducting this type of research.

Reviewer 2 (Anonymous)
2. The Introduction section is adequate concerning the quoted literature. However, I invite the authors to better explain in the Intro section why a new study was needed – what is the knowledge gap - considering the high-quality systematic review and meta-regression of Carter and Worthington in 2015. Their analysis included 24 studies on Europeans, so this constituted a large sample size for meta-analysis.
Authors’ Response: The study of Carter and Worthington had a large sample size and a valid methodology, and so we used the reported value for third molar agenesis as a reference value for our sample. However, Carter and Worthington (2015) looked at the prevalence of third molar agenesis in the population. This meta-analysis did not assess whether other teeth were missing or not. In our study, we investigated the chances of a missing third molar when other teeth are also missing. The last part of the introduction has been revised to clarify this issue.
This partially addresses this concern raised.

Experimental design
5. The inclusion criteria for the controls could be described better. It is only stated that a control group of 303 individuals without agenesis of teeth other than third molars, were matched for age and sex, but what about the other criteria as for the cases group?
Authors’ Response: Except from tooth agenesis, the other inclusion criteria of the control group were identical to those of the agenesis group. This is now clearly stated in the Methods section.
--This is now adequately addressed.

Validity of the findings
9. When considering the percentages of the number of missing third molars per individual in the control and the agenesis group, a tendency was found towards more missing third molars in the agenesis group. This is the key finding of the study. The authors only mention ‘’genetic factors’’ which is a very vague term. The discussion could be enhanced by trying to explain this finding from the embryological viewpoint, f.e. the studies of Inger Kjaer in Copenhagen.
Authors’ Response: We thank the reviewer for the suggestion. In terms of developmental timing, the third molar is the last tooth in the molar series and also the last tooth to develop in the dentition. It is known that the last tooth in each tooth series shows more often developmental disturbances, including agenesis, thus being more vulnerable to genetic or environmental factors that might be present (Townsend et al. Morphogenetic fields within the human dentition: A new: Clinically relevant synthesis of an old concept. Arch Oral Biol. 2009;54:S34–S44. & Gkantidis et al. Patterns of non-syndromic permanent tooth agenesis in a large orthodontic population. Arch Oral Biol. 2017;79:42-47.) This might be another contributing factor relevant to the present results. It has also been shown that overall dental development is delayed in patients with tooth agenesis, compared to controls, with a weak correlation between dental developmental stage and number of missing teeth (Lebbe et al. 2017). This information has been now added in the discussion section.
--This is a much needed improvement to the discussion.

Reviewer 2 ·

Basic reporting

The manuscript (revision v2) is clearly written in professional English and has been improved according to the referees’ comments.
The Intro introduces the reader to the topic and literature cited is adequate.
Table legends have been improved.

Experimental design

Based on the comments of the referees, in the revised version the authors have adequately addressed issues that were raised.

Validity of the findings

I would not characterize this study as novel, but data is valid and could be useful in further studies. I have nothing to add here.

Additional comments

The authors have answered the comments of the referees satisfactorily and revised the manuscript accordingly. I have no further comments or suggestions for the authors.

---

## Round 0.3 · accepted · Accept

The revision addresses the reviewer's concerns. I don"t see a reason for sending it for another round of review.